# Periodontal Disease and Nonsurgical Periodontal Therapy on the OHRQoL of the Patient: A Pilot Study of Case Series

**DOI:** 10.3390/dj11040094

**Published:** 2023-04-03

**Authors:** María José Moya-Villaescusa, Arturo Sánchez-Pérez, Jesús Esparza-Marín, Alfonso Jornet-García, José María Montoya-Carralero

**Affiliations:** Department of Periodontology, Faculty of Medicine and Dentistry, University of Murcia, 30008 Murcia, Spain

**Keywords:** nonsurgical periodontal therapy, periodontitis, quality of life, scaling, subgingival, OHIP

## Abstract

The objective of this prospective study was to assess oral health-related quality of life (OHR-QoL) in patients with periodontitis and nonsurgical periodontal treatment. A prospective, longitudinal case series study was carried out at the University Dental Clinic of the Morales Meseguer Hospital in Murcia. Eighty periodontal patients with periodontitis, aged 22 to 72 years, were included in this study. The Spanish version of the Oral Health Impact Profile 14 (OHIP-14sp) questionnaire was used on two occasions: at the first visit and one month after completing the nonsurgical periodontal treatment. Clinical measurements of probing depth, plaque index and bleeding on probing were performed at baseline and after periodontal therapy. Data were analyzed using Student’s t-test and analysis of variance. We found statistically significant differences (*p* < 0.05) between the pre- and post-treatment in all the dimensions studied except disability. Similarly, statistically significant differences were also found according to the degree or stage of the disease. However, these differences were not observed with respect to the extent of periodontal disease. Periodontal disease has a negative impact on the OHRQoL of periodontal patients, especially in severe stages (III–IV). Basic periodontal treatment can improve the OHRQoL of periodontal patients one month after treatment.

## 1. Introduction

Periodontitis is a chronic, infectious and inflammatory disease that irreversibly affects the supporting structures of the tooth, which can lead to tooth loss. According to the Global Burden of Disease Study (GBD), it is the sixth most prevalent disease in humanity [1]. Furthermore, worldwide, 11% of the population has severe periodontal disease, which is one of the causes of 20% of adults over 60 being edentulous [1]. The high prevalence of periodontal disease produces an important public health problem, causing tooth loss. It involves chewing, phonetic, psychological and social problems that affect the well-being and quality of life of the patient.

In general, during its initial stages, periodontal disease is not usually associated with clinical symptoms perceived by the affected individual. However, in the long term, it can cause signs and symptoms with a cumulative effect [2]. These signs and symptoms have a significant impact on patients in terms of severity, frequency and duration [3]. The improvement of these symptoms and, thereby, the improvement of the patient’s individual level of oral well-being is one desirable outcome of periodontal therapy [4]. However, for many years, we have examined, diagnosed and treated periodontal patients without knowing their opinion, that is, how their daily life was affected. In general, greater importance has been given to objectively measurable variables such as plaque index, pocket probing depth (PPD), bleeding on probing (BOP), clinical insertion level (CAL) or bone level (BL) than to the patient’s subjective perception.

Increasing life expectancy and improving quality of life are the two goals of the 21st century. Quality of life is defined by the WHO as the individual perception of life in the context of the culture and value system in which they live and in relation to their goals, expectations, standards and concerns.

In all fields of health sciences, and particularly in dentistry, this treatment-oriented approach and the perceptions that the patient may experience are of paramount importance. Dentists should be aware of the patient’s perceptions, both physical and psychological. With this knowledge, the treatment plan can be personalized and thus achieve greater patient adherence and motivation.

When assessing the quality of life of periodontal patients, the interpretation of the clinical examination through periodontal diagnostic indices is not useful. These indices do not reflect the limitations that patients suffer when they carry out their daily oral activities [5].

To achieve this purpose, oral epidemiology has implemented the use of questionnaires called Oral Health-Related Quality of Life (OHRQoL) [6], which evaluate the impact of oral health and dental interventions on the quality of life of individuals. These questionnaires try to objectify and give value to the experiences and perceptions of the patients regarding the progress of their oral health, registering their improvement or deterioration. To do this, the person surveyed must indicate their agreement or disagreement about a statement or item, which is done through an ordered and one-dimensional scale [7].

The scores obtained are assessed twice, before and after any treatment is performed. The effect of treatment on the outcome can be quantified as the difference between the two scores [8].

The most widely used OHRQoL questionnaire is the “Oral Health Impact Profile” (OHIP). Its simplified version is the OHIP-14 [9]. The OHIP-14sp is the Spanish version of the OHIP-14 questionnaire and was validated in Spain by Montero-Martín et al. [5].

The intention of our study was to determine the impact that non-surgical treatment has on the QHRoL of patients who are seen in a university setting by undergraduate students. We also tried to collect the effect of the treatment received (PROMs) on the patients and to evaluate the ability to motivate and continue with the treatment protocol.

On the other hand, we want to test whether the effect of non-surgical treatment on patients in southern European countries differs between regions.

The patients included in this study followed the recommendations of the European Federation of Periodontology (EFP) S3 treatment guide. In summary: patient motivation, oral hygiene instructions, supragingival plaque removal and subgingival plaque instrumentation (steps 1 and 2 of the guideline) were performed.

The general objective of this study was to evaluate the early post-treatment impact of nonsurgical periodontal treatment on the quality of life of patients. As secondary objectives, we intend to determine whether variables such as stage, grade and extent of periodontal disease can affect OHRQoL in a Spanish population.

## 2. Materials and Methods

### 2.1. Study Population

The present cohort study was comprised of 80 consecutive patients diagnosed with periodontitis at any stage (I–IV) and with any grade (A, B or C). The study period was from September 2021 to May 2022, according to the clinical practices of the corresponding academic year. The sample was selected following a nonprobabilistic sequential model, that is, as they attended the University Dental Clinic of the Morales Meseguer Hospital in Murcia.

Patients who agreed to be included in this study completed the OHIP-14sp, which is an abbreviated version of the original 49-question OHIP [6]. The OHIP-14sp consists of 14 questions on the frequency of discomfort caused by oral conditions and is grouped into 7 domains with 2 subdomains in each question:Functional limitation (trouble in pronouncing words, worsened sense of taste),Handicap (life in general is less satisfying, totally unable to function),Psychological disability (difficult to relax, being slightly embarrassed),Psychological discomfort (having felt tense, being self-conscious),Physical disability (having to interrupt meals, diet has been unsatisfactory),Physical pain (uncomfortable eating any foods, painful aching)Social disability (irritable with other people, difficulty performing usual jobs).

The answers are recorded on a Likert scale, with values ranging from 0 to 4 coded as follows: 0 ‘never’, 1 ‘hardly ever’, 2 ‘occasionally’, 3 ‘fairly often’ or 4 ‘very often’. Responses of the OHIP-14sp are summed to give the total OHIP score and can range from 0 to 56, with a high score indicating a worse OHRQoL. The values can also be summed within the seven subdomains.

For the design of this study, we followed the methodology proposed by Theodoridis for the countries of southern Europe to be able to compare our results [10].

### 2.2. Ethical Considerations

The study was approved by the research ethics committee of the University of Murcia; ID: 3354/2021; 30 June 2021. The work described has been carried out in accordance with The Code of Ethics of the World Medical Association (Declaration of Helsinki, 1964) for experiments involving humans. This is a prospective, longitudinal case series study. All patients were informed about the possible risks and benefits of the study and gave their written informed consent.

Inclusion criteria included: (1) patients with the presence of pockets ≥ 4 mm with bleeding on probing or presence of deep periodontal pockets [≥6 mm]., (2) being over 18 years old, (3) having signed the informed consent, (4) having at least 15 teeth in the mouth and (5) having completed the basic periodontal treatment and coming one month after the last scaling and root planing. (6) Patients had to understand and correctly complete the OHIP-14sp questionnaire.

The exclusion criteria were as follows: (1) pregnancy or breastfeeding; (2) systemic disease that could modify the immune response; (3) history of nonsurgical or surgical periodontal treatment in the previous 12 months and (4) history of neoplasm or maxillofacial radiotherapy in the previous 5 years.

### 2.3. Procedures

The clinical procedure was as follows: once the patient went to the Dental Clinic, the students filled in the periodontal medical history in all its sections and diagnosed the patient as healthy or with gingivitis or periodontitis with their stages, degrees and extent [11]. In the case of periodontitis, the patient was informed of the necessary treatment and the study that we intended to perform. If the patient agreed to participate in the study and underwent nonsurgical periodontal treatment (steps 1 and 2 of the EFP S3 guideline), they received some information sheets and signed the informed consent form. Then, they were asked about the items of the first OHIP-14sp questionnaire (pretreatment questionnaire).

The recommendations of the EFP guidelines for the treatment of periodontitis stages I–III were followed. The patients were informed of their condition, instructed in oral hygiene techniques and advised on the abandonment of harmful habits (smoking). Subsequently, supragingival cleaning was performed by ultrasound, and where pockets larger than 4 mm with bleeding or pockets larger than 6 mm persisted, manual subgingival instrumentation was performed using a reduced set of Gracey curettes. The second OHIP-14sp questionnaire (post-treatment questionnaire) was completed for the patient one month after the last scaling and root planing during their periodontal re-evaluation (Figure 1).

At the end of the study period, appropriate further periodontal treatment, including periodontal surgery, was prescribed for those subjects who still had sites with residual probing more than 6 mm deep and bleeding upon probing.

### 2.4. Statistical Analysis

The sample size was determined by previous studies [10,12,13], as well as the sample size calculator provided by Elsevier through their primary care support (https://www.fisterra.com/formacion/metodologia-investigacion/determinacion-tamano-muestral/#sec4 (accessed on 1 September 2021)). The variable used to calculate the sample size was the OHIP-14 for a precision of 1.5 points of difference. More specifically, for α = 0.05 and a power of 80%, a total sample minimum of 65 patients was needed. An extra 15% was added to this sample of 65 patients, resulting in a final sample size of 76 individuals. Our sample size agrees with that obtained by Theodoridis et al. [10] for a Greek population.

Data analysis was conducted in the Statistical Package for the Social Sciences (SPSS) program, version 23.0 (SPSS, Chicago, IL, USA). All outcome variables were adjusted for normality according to the Kolmogorov–Smirnov test as well as the Shapiro–Wilk test. Descriptive and inferential statistics were analyzed using Tukey’s exploratory data analysis and Student’s *t*-test for the 7 variables measured in the questionnaires: functional limitation, physical pain, psychological discomfort, physical, psychological and social disability, and handicap, as well as for the total scores OHIP 14-sp pre- and post-treatment. To assess the influence of stage, grade and extension on OHRQoL, one-factor analysis of variance (ANOVA) was used. The level of significance was *p* < 0.05, and the mean was adjusted at 5% and 95% confidence intervals.

## 3. Results

There were no dropouts during the study, and all 80 patients completed the re-evaluation at one month.

A total of 80 periodontal patients, 43 men and 37 women were included in this study. The mean age of the participants was 51.21 ± 12.91 years (median: 54 years; range: 22–72 years), and the mean number of teeth was 25.69 ± 0.70 (median: 26; range: 15–30).

In total, 8 of these patients (10.25%) had between 15 and 20 teeth in the mouth, twenty-four (30.77%) had between 21 and 25 teeth, and most of the patients (48; 61.53%) had between 26 and 32 teeth in their mouth. Regarding the extent of periodontitis, in most cases, it was generalized (68.8%). Regarding stages and grades, 9 patients (11.3%) had stage I, 15 patients had stage II (18.8%), 36 patients (45%) had stage III and 18 patients (22.5%) had stage IV. Further, 14 patients (17.5%) had grade A, 51 patients (63.7%) had grade B and 13 patients (16.3%) had grade C. In two cases, the stage and grade were not recorded.

Before treatment, the mean total OHIP-14sp summary score for the whole study cohort was 11.70 ± 8.38 (median: 10; range: 0–38). In total, 2 patients had a summary score of 0, representing 2.5% of the whole study cohort.

After treatment, the mean total OHIP-14sp summary score for the whole study cohort was 9.37 ± 8.16 (median: 8; range: 0–34). Similar to before treatment, 2 patients had a summary score of 0, representing 2.5% of the whole study cohort. Table 1 summarizes the scores of each dimension (functional limitation, physical pain, psychological discomfort, physical, psychological and social disability, and handicap) of the OHIP-14sp pre- and posttreatment questionnaires (Table 1).

Comparing the different dimensions studied in the pre- and post-treatment questionnaires, we found statistically significant differences between all of them, except for “handicap” (Table 1).

Regarding the variables “grade”, “extension” and “stage” of periodontal disease, we found statistically significant differences in grade (Table 2) and stage (Table 3) but not in extension.

The patient-reported outcome measure (PROM) has been highlighted as a variable of paramount importance in patients’ daily lives above surrogate variables such as PPD, CAL, BOD or BL [2,3,14,15,16]. However, very few clinical studies refer to this important variable.

In order to address the patient’s point of view and receive his or her impressions, preferences and limitations, several tools have been developed for use in dentistry. Among them, the OHIP, in its different versions validated for each country, is a reliable approach to the changes made in the oral environment by any procedure performed. Both OHIP-49 and OHIP-49 have been widely used to assess OHRQoL. However, the OHIP-14 is often preferred over the OHIP-49 because of its greater ease, fewer questions and its reliability, validity and accuracy (Figure 2).

In both cases, before and after periodontal treatment, the dimension that most affected the periodontal patient was “physical pain”, translated as tooth pain or sensitivity. Second, the “psychological discomfort” manifested by being unhappy with the appearance of their teeth. The third place is occupied by the “functional limitation” that can cause the loss of teeth due to periodontal disease. This leads to poor chewing and, consequently, digestive discomfort if the patient does not replace them with a prosthesis.

Although the periodontal disease does not usually cause acute pain, except in necrotizing diseases and periodontal abscesses, sensitivity and chewing disability are frequent. Periodontal attachment loss and periodontal treatment lead to recessions, where the root of the tooth is exposed [17], causing sensitivity. In addition, on an aesthetic level, recession gives rise to a long tooth, with loss of papillae and black interdental triangles, which produces psychological discomfort in the patient [18].

These results agree with those obtained by Habashneh et al. [19], where the item with the highest score was also physical pain.

Our results are also in agreement with those classically obtained with nonsurgical treatment [20,21] and are in agreement with the review by Shanbhag et al. [22]. The latter authors also found that surgical treatment, paradoxically, has a small impact on OHRQoL. We believe that a possible explanation for this lack of improvement is that patients requiring surgical treatment are in the advanced stages of the disease with an irreversible loss of insertion that may not be recoverable.

Regarding the “handicap” dimension, in the sense of being unable to function or work due to dental problems, in both the pre- and post-treatment questionnaires, the values were very low, with no statistically significant differences. This means that in no case a periodontal disease or its treatment prevents the patient from carrying out their daily work. It does not affect, from a global point of view, the score obtained in the OHRQoL.

However, we would like to highlight this aspect of our study, the persistence of the handicap dimension. Which we believe is related to the irreversibility of periodontitis. Patients improve in the other components and may be relieved of their symptoms, but functionally they suffer the consequences of the loss of tooth support. Similarly, the Theodoridis study also found an absence of improvement in physical and psychological disabilities [10], which is consistent with the patient’s perceived insecurity about their periodontitis and also reflects the irreversibility of the disease in advanced stages.

In contrast, total OHIP 14-sp scores were significantly reduced after periodontal treatment, which means that periodontal patients improved their OHRQoL after nonsurgical periodontal treatment. These results are in agreement with those obtained by Goel and Baral [23], Mendez et al. [24] and Botelho et al. [25].

If we analyze the results obtained, taking into account the degree of periodontal disease, we do not find statistically significant differences between patients with different degrees of periodontal disease, except for post-treatment “physical pain”. The impact of the “physical pain” dimension in patients with grade C periodontitis was significantly greater than that in patients with grade A periodontitis. This could be explained because grade C patients have a higher risk of progression of periodontal disease and respond worse to periodontal treatment than grade A patients [26]. This may make the “physical pain” dimension not improve after periodontal treatment in grade C periodontitis patients.

Goergen found a correlation similar to ours [27], where nonperiodontal or stage I grade B patients obtained low scores, while stage II grade C patients or stage III/IV grade B or C patients had the highest OHIP-14 scores.

Regarding the extent of periodontal disease, we did not find statistically significant differences between the OHRQoL of patients with localized and generalized periodontitis, either before or after basic periodontal treatment. However, we found differences when we took into account the stage or severity of the periodontal disease.

Pretreatment patients diagnosed with severe periodontal disease (stage IV) had significantly higher values than those diagnosed with stages I, II and III in the “physical disability” and “social disability” dimensions. It can be interpreted that only in the severe stages of periodontal disease do patients perceive a certain disability in speech that affects their social life. In the “psychological disability” dimension, patients with stage IV showed values significantly higher than those with stage III. This means that the patient is in the most severe stage of periodontal disease when their OHRQoL begins to be affected psychologically. In the “handicap” dimension, patients with stage IV registered significantly higher figures than those with stages I and II. We found that in the most advanced forms of periodontal disease, patients may be unable to carry out their work in the usual way. These differences mean that the total pretreatment OHIP-14sp scores of patients with severe periodontitis are significantly higher than those of patients with mild periodontitis; that is, they have a worse OHRQoL.

After nonsurgical periodontal treatment, these differences between the stages of periodontal disease also appear in stage IV but only in the dimensions of “psychological discomfort” and “handicap”. Our interpretation of these results is that nonsurgical periodontal treatment will not improve the aesthetics of the patient or their chewing function due to advanced insertion loss or the lack of teeth. However, this does not affect the total score of post-treatment OHIP-14sp.

These results are similar to those obtained by Karaaslan and Dikilitaş [28] and Habashneh et al. [19]. In contrast, Ustaoğlu et al. [29] found that “functional limitation” and “social disability” had a greater impact on patients with severe periodontitis than on the rest of the groups. They are also in line with those carried out in Greece, which represents a similar Mediterranean framework [10]. In our view, QoL should be evaluated under the concept provided by the WHO, which implies incorporating the patient’s culture and environmental factors. This last aspect is influenced by geographical and cultural aspects.

This study has a number of limitations. One of the limitations of this study is the short follow-up period of the patients, which was only 1 month after the last scaling and root planing. Sometimes, especially in patients with severe periodontitis, the healing time of the tissues and, therefore, the time to notice changes in their oral health is usually longer. Although in advanced stages of periodontitis, periods of 6 to 8 weeks are recommended for re-evaluation, in our study, we tried to maintain a re-evaluation protocol of only 1 month for different reasons.

First, once patients have undergone non-surgical periodontal treatment, they suffer a significant dispersion. Some patients improve notably and only have to undergo maintenance (the minority). Other patients need to repeat the non-surgical treatment due to persistent inflammation or pockets larger than 4 mm with bleeding (the majority). 

Finally, even if some of them are scheduled for a second non-surgical session, a variable percentage will end up needing surgical treatment, which implies longer re-evaluation times and a greater dispersion depending on the surgical technique used.

Likewise, our intention was to compare our results with those obtained by Theodoridis et al. regarding their evaluation after non-surgical periodontal treatment in a southern European setting. This author also evaluates the outcome of undergoing periodontal surgery, finding that it did not bring any significant change with respect to OHR-QoL.

Another limitation was that patients who come to the University Dental Clinic belong to a specific population group. This fact could imply the nongeneralization of our results to the general population. Due to the fact that the patients previously go through a general diagnostic and screening phase in the dental clinic, most of the patients who come to the periodontics unit are aware of their condition, with a high degree of compliance that is emphasized by our students. For this reason, the percentage of patients who agreed to participate in the study was 100%. We are aware that this percentage remained constant due, among other aspects, to the personalized follow-up carried out by the students and the short follow-up period of the study (2 months).

Finally, the patients who come to the dental clinic are part of a multidisciplinary treatment, where some of them only need periodontal treatment, and others need a combination of other treatments (conservative, orthodontics, prosthesis, implants, etc.). The vision and importance of this integration are enhanced by the collaboration between the different units as well as the integration that takes place in the units of integrated treatment of adult, pediatric and special needs patients.

For all these reasons, this study should be considered as a pilot study with preliminary results.

Possible future lines of research could be oriented towards the conformation of a map of PROMs in the non-surgical treatment of Southern European countries. Another interesting line of research is the 6-month or 12-month follow-up on the degree of improvement observed by OHIP-14 with respect to the baseline score.

Likewise, it would be of great interest to continue the study with respect to the outcome variables (BL, PPD, BOP or CAL) and their possible relationship with respect to the degree of patient satisfaction.

In conclusion, periodontal disease has a negative impact on OHRQoL in periodontal patients, especially in severe stages. Basic periodontal treatment has a positive impact on the OHRQoL of periodontal patients one month after treatment. The degree of periodontal disease influences post-treatment physical pain, while the extent of periodontal disease does not seem to affect OHRQoL.

## Figures and Tables

**Figure 1 dentistry-11-00094-f001:**
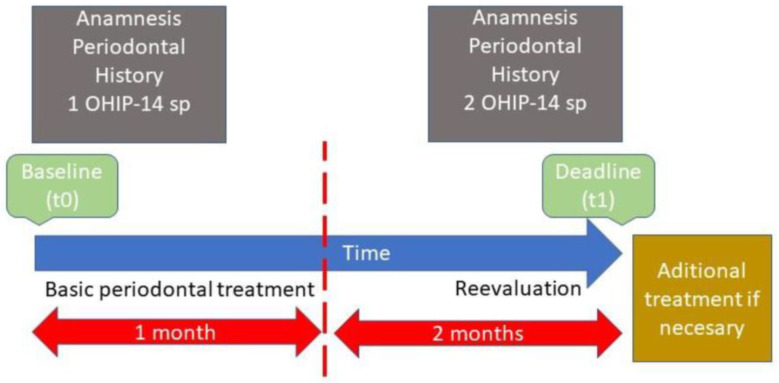
Study flow chart.

**Figure 2 dentistry-11-00094-f002:**
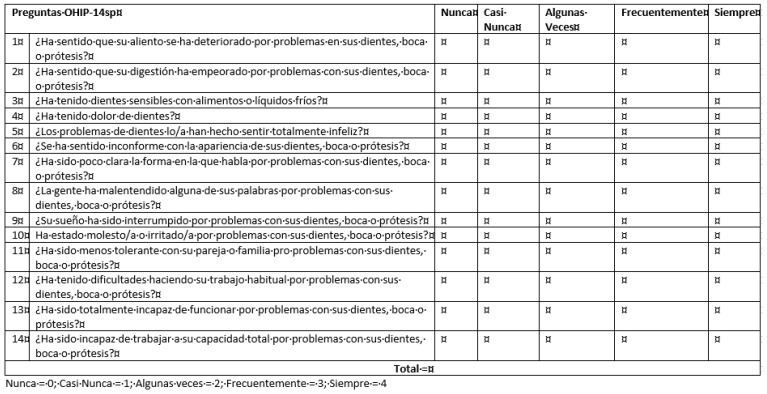
Spanish version of the oral health impact profile 14 questionnaire (OHIP-14sp).

**Table 1 dentistry-11-00094-t001:** Mean and standard deviation with respect to the 7 dimensions of OHIP-14sp pre- (OHIP-14sp1) and post-treatment (OHIP-14sp2) and the total score.

Variables	Mean (SD)	Mean (SD)	Matched Differences				
	OHIP-14sp1	OHIP-14sp2	Mean	SE	95% CI		*p*
Functional limitation	2.13 (1.76)	1.55 (1.71)	0.57	0.12	0.32	0.83	**
Physical pain	3.61 (1.89)	3.09 (1.70)	0.52	0.17	0.18	0.87	**
Psychological discomfort	2.73 (2.51)	2.25 (2.49)	0.47	0.16	0.14	0.80	**
Physical disability	0.84 (1.58)	0.60 (1.31)	0.23	0.11	0.01	0.46	*
Psychological disability	1.43 (1.82)	1.09 (1.65)	0.33	0.13	0.06	0.61	*
Social disability	0.64 (1.20)	0.45 (0.99)	0.18	0.08	0.02	0.34	*
Handicap	0.44 (1.11)	0.34 (1.02)	0.10	0.06	−0.03	0.23	0.13
Total OHIP-14sp	11.70 (8.38)	9.38 (8.16)	2.32	0.39	1.54	3.10	**

SD: standard deviation; SE: standard error; CI: confidence interval; * *p* < 0.05; ** *p* < 0.01.

**Table 2 dentistry-11-00094-t002:** ANOVA of the significant dependent variables (post-treatment: 2) with respect to the grade of periodontal disease (A, C).

Dependent Variables	Grade	Grade	DM	SE	95% CI		*p*
Physical pain 2	A	B	−0.943	0.067	−1.95	−0.07	0.067
		C	−1.401	0.034	−2.69	−0.11	0.034 *

DM: Difference of mean; SE: Standard error; CI: Confidence interval; * *p* < 0.05.

**Table 3 dentistry-11-00094-t003:** ANOVA of the significant dependent variables (pretreatment:1 and post-treatment: 2) with respect to the stage of periodontal disease (I, II, III, IV).

Dependent Variables	Stage	Stage	DM	SE	95% CI		*p*
Psychological discomfort1 (Pretreatment)	III	IV	−1.55	0.71	−2.97	−0.14	*
Physical disability1 (Pretreatment)	IV	I	1.27	0.62	0.03	2,53	*
		II	1.45	0.53	0.38	2.53	**
		III	0.94	0.44	0.06	1.83	*
Psychological disability1 (Pretreatment)	II	III	−1.39	0.54	−2.49	−0.30	*
Social disability1 (Pretreatment)	IV	I	1.11	0.47	0.16	2.06	*
		II	1.13	0.40	0.32	1.95	**
		III	0.72	0.33	0.05	1.39	*
Handicap1 (Pretreatment)	IV	I	0.94	0.44	0.06	1.82	*
		II	0.87	0.37	0.12	1.63	*
Psychological discomfort2 (Post-treatment)	IV	I	2.00	1.00	0.01	3.99	*
		III	1.55	0.70	0.15	2.97	*
Handicap2 (Post-treatment)	IV	II	0.72	0.35	0.02	1.42	*
Total OHIP-14sp1 (Pretreatment)	II	IV	−6.18	2.89	−11.95	−0.42	*

DM: Difference of mean; SE: Standard error; CI: Confidence interval; * *p* < 0.05; ** *p* < 0.014. Discussion.

## Data Availability

Data is available from the date the study is accepted for publication and may be requested from the corresponding author by e-mail: arturosa@um.es.

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
