# Peer review of "Periodontal Disease and Nonsurgical Periodontal Therapy on the OHRQoL of the Patient: A Pilot Study of Case Series"

_dentistry, 2023, doi:10.3390/dj11040094_

Round 1

Reviewer 1 Report

The article is simple and clear, it takes into account only 80 patients during a short period.

The results are very predictable. The second questionary is given just when a reevaluation can be performed but not when it is actually recommended for a stage III or IV, which should be performed 6-8 weeks after scaling and root planing. It is recommended to give the second or a third questionary 3-6  months after scaling and root planning when clinical changes are notorious for the clinician and the patient, this way the answers will have more value.

The greatest change in probing depth reduction and gain in clinical attachment occurs within 1–3 months post-scaling and root planing, although healing and maturation of the periodontium may occur over the following 9–12 months. (Cobb CM. Clinical significance of non-surgical periodontal therapy: an evidence-based perspective of scaling and root planing. Journal of clinical periodontology. 2002 May;29:22-32.)

Author Response

Referee 1

The article is simple and clear, it takes into account only 80 patients during a short period.

The results are very predictable. The second questionary is given just when a reevaluation can be performed but not when it is actually recommended for a stage III or IV, which should be performed 6-8 weeks after scaling and root planing. It is recommended to give the second or a third questionary 3-6  months after scaling and root planning when clinical changes are notorious for the clinician and the patient, this way the answers will have more value.

The greatest change in probing depth reduction and gain in clinical attachment occurs within 1–3 months post-scaling and root planing, although healing and maturation of the periodontium may occur over the following 9–12 months. (Cobb CM. Clinical significance of non-surgical periodontal therapy: an evidence-based perspective of scaling and root planing. Journal of clinical periodontology. 2002 May;29:22-32.)

We agree with the reviewer given that the remodeling time, as he rightly points out, can extend over several months and last up to 2 years. In this period the acycatrization goes through a first inflammatory phase (from days to weeks), a proiferative phase (from weeks to months) and a remodeling phase (from months to years).

We have added this entire paragraph in the discussion, within the limitations of the study.

“Although in advanced stages of periodontitis, periods of 6 to 8 weeks are recommended for re-evaluation, in our study, we tried to maintain a re-evaluation protocol of only 1 month for different reasons:

First, because of the length of the academic year, since treatments and reassessment are performed during the academic year, it is not possible to perform follow-ups longer than 1 month after the last non-surgical treatment. This is because students spend the first term acquiring basic skills.

Secondly, once patients have undergone non-surgical periodontal treatment, they suffer a significant dispersion. Some patients improve notably and only have to undergo maintenance (the minority). Other patients need to repeat the non-surgical treatment due to persistent inflammation or pockets larger than 5 mm with bleeding (the majority).

Finally, even if some of them are scheduled for a second non-surgical session, a variable percentage will end up needing surgical treatment, which implies longer re-evaluation times and a greater dispersion depending on the surgical technique used.

Likewise, our intention was to compare our results with those obtained by Theodoridis et al. regarding their evaluation after non-surgical periodontal treatment in a southern European setting. This author also evaluates the outcome of undergoing periodontal surgery, finding that it did not bring any significant change with respect to OHR-QoL.”

We would like to thank the referee for all the time and effort spent in supervising our work. We hope that this new corrected version will be considered appropriate for publication.

Sincerely yours.

Reviewer 2 Report

It is an interesting follow-up investigation, but needs improvement in some aspects. 

In materials and methods :

*Clarify who performed the clinical inspection and whether it was the same operator as in the reevaluation. Indicate if there was performed any calibration  

*Mention the periodontal classification used for the diagnosis. 

In results : Indicate the acceptance rate 

In discussion, mention how this quality of life could be improved in periodontal patients. Emphasise the need for multidisciplinary work. 

Author Response

Referee 2

Comments and Suggestions for Authors

It is an interesting follow-up investigation, but needs improvement in some aspects. 

 In materials and methods:

*Clarify who performed the clinical inspection and whether it was the same operator as in the reevaluation. Indicate if there was performed any calibration  

*Mention the periodontal classification used for the diagnosis. 

In results : Indicate the acceptance rate 

In discussion, mention how this quality of life could be improved in periodontal patients. Emphasise the need for multidisciplinary work. 

  1. As requested by the referee, we have added the following paragraph in the materials and methods: "The first evaluation and the final reevaluation was always performed by the same examiner, under the supervision of his tutor. The evaluators all underwent a calibration process during the first four-month period of the course, during which they acquired the necessary skills to be able to treat the patients".

  1. Although we refer to the classification used on page 3 lines 123 reference 11, we agree with the referee and have expressly referred to the classification used by adding the following paragraph: "To classify our patients, we used the current 2018 classification [11]".

  1. As requested by the referee, we have added the following paragraph in the discusion “Other limitation was that, patients who come to the University Dental Clinic belong to a specific population group. This fact could imply nongeneralization of our re-sults to the general population. Due to the fact that the patients previously go through a general diagnostic and screening phase in the dental clinic, most of the patients who come to the periodontics unit aware of their condition, with a high degree of compliance that is emphasized by our students. For this reason, the percentage of patients who agreed to participate in the study was 100%. We are aware that this percentage remained constant due, among other aspects, to the personalized follow-up carried out by the students and the short follow-up period of the study (2 months).”

  1. As pointed out by the referee, we have added the following paragraph to improve our text: "Finally, the patients who come to the dental clinic are part of a multidisciplinary treatment, where some of them only need periodontal treatment and others need a combination of other treatments (conservative, orthodontics, prosthesis, implants, etc.). The vision and importance of this integration is enhanced by the collaboration between the different units as well as the integration that takes place in the units of integrated treatment of adult, pediatric and special needs patients".

We would like to thank the referee for all the time and effort spent in supervising our work. We hope that this new corrected version will be considered appropriate for publication.

Sincerely yours.

Reviewer 3 Report

The present manuscript aims at evaluating the effect of initial periodontal treatment on the oral health-related quality of life of affected patients. Overall, the work has some merits, since the literature is particularly lacking of studies assessing PROMs. However, several reporting and methods issues have to be fixed in order to improve the overall quality. 

Intro

- From the introduction, it is not clear what the present study adds to the literature. Please, elaborate. 

- The author refer to the treatment as 'basic periodontal treatment'. Please, refer to the term introduced by recent EFP guideline (PMID: 32383274).

M&M

- What do the authors mean by chronic periodontal patients? Also, in inclusion criteria what does it mean 'being a periodontal'??

- 'The treatment protocol was based on hygiene techniques and scaling and root planing'. Please, refer more specifically to the treatment procedures employed (PMID: 31889320). 

- Which was the primary outcome selected for sample size calculation? (PMID: 31661431) Please, explicit whether the data followed a normal distribution and specify that you used a statistical test for paired samples. Be aware that student t test can not be applied for non parametric data that you seemed to obtain. 

Results and conclusions

- There is a complete lack of reporting for variables assessing the efficacy of periodontal treatment. To this regard, it would be important to discuss results on the number of residual pockets after therapy (PMID: 34517433) since this value may better reflect the burden of periodontitis as well as the need for additional therapy than mean parameters. 

- Tables 2 and 3 are very difficult to follow. Please, make an effort to make them more informative. 

- In the discussion, explaining again how OHIP questionnaire works is pleonastic. 

- The last part of the discussion may benefit from some future perspectives with the aim of replicating this study in patients with systemic conditions such as diabetes in which the impact of periodontal treatment on OHRQL may be larger (PMID: 33924022).

Author Response

Referee 3

Comments and Suggestions for Authors

The present manuscript aims at evaluating the effect of initial periodontal treatment on the oral health-related quality of life of affected patients. Overall, the work has some merits, since the literature is particularly lacking of studies assessing PROMs. However, several reporting and methods issues have to be fixed in order to improve the overall quality. 

 Intro

1- From the introduction, it is not clear what the present study adds to the literature. Please, elaborate. 

2- The author refer to the treatment as 'basic periodontal treatment'. Please, refer to the term introduced by recent EFP guideline (PMID: 32383274).

M&M

3- What do the authors mean by chronic periodontal patients? Also, in inclusion criteria what does it mean 'being a periodontal'??

4- 'The treatment protocol was based on hygiene techniques and scaling and root planing'. Please, refer more specifically to the treatment procedures employed (PMID: 31889320). 

5- Which was the primary outcome selected for sample size calculation? (PMID: 31661431) Please, explicit whether the data followed a normal distribution and specify that you used a statistical test for paired samples. Be aware that student t test can not be applied for non parametric data that you seemed to obtain. 

Results and conclusions

6- There is a complete lack of reporting for variables assessing the efficacy of periodontal treatment. To this regard, it would be important to discuss results on the number of residual pockets after therapy (PMID: 34517433) since this value may better reflect the burden of periodontitis as well as the need for additional therapy than mean parameters. 

7- Tables 2 and 3 are very difficult to follow. Please, make an effort to make them more informative. 

8- In the discussion, explaining again how OHIP questionnaire works is pleonastic. 

9- The last part of the discussion may benefit from some future perspectives with the aim of replicating this study in patients with systemic conditions such as diabetes in which the impact of periodontal treatment on OHRQL may be larger (PMID: 33924022).

1.- As suggested by the referee, we have expanded the reasons that led us to conduct this study: “The intention of our study was to determine the impact that non-surgical treatment has on the QHol of patients who are seen in a university setting by undergraduate students. We also tried to collect the effect of the treatment received (PROMs) on the patients and to evaluate the ability to motivate and continue with the treatment protocol.

On the other hand, we want to test whether the effect of non-surgical treatment on patients in southern European countries differs between regions.”

2.- As suggested by the referee, we have included the following paragraph, where we explain our adherence to the EFP S3 guidelines:

“The patients included in this study followed the recommendations of the European Federation of Periodontology (EFP) S3 treatment guide. In summary: patient motivation, oral hygiene instructions, supragingival plaque removal and subgingival plaque instrumentation (steps 1 and 2 of the guideline) were performed.”

3.- We agree with the referee and have specified the terms " chronic periodontal patients " and "'being a periodontal'??..." by substituting the following terms:

 “patients diagnosed with periodontitis at any stage (I-IV) and with any grade (A, B OR C).”

Patients with presence of pockets ≥4 mm with bleeding on probing or presence of deep periodontal pockets [≥6 mm].”

4.- We agree with the referee's suggestion and have included the following paragraph explaining the treatment protocol followed:

“The recommendations of the EFP guidelines for the treatment of periodontitis stages I-III were followed. Basically, the patients were informed of their condition, instructed in oral hygiene techniques and advised on the abandonment of harmful habits (smoking). Subsequently, supragingival cleaning was performed by ultrasound and where pockets larger than 4 mm with bleeding or pockets larger than 6 mm persisted, manual subgingival instrumentation was performed using a reduced set of Gracey curettes.”

5.- We have included the following clarification on the sample size calculation and the variable used. This information has been expanded on page 4, lines 139-143: references 10, 12 y 13 (PMID: 32466149, PMID: 17722435, PMID: 22092418).

“The sample size was determined by previous studies [10, 12,13]. As well as the sam-ple size calculator provided by Elsevier through their primary care support (https://www.fisterra.com/formacion/metodologia-investigacion/determinacion-tamano-muestral/#sec4). The variable used to calculate the sample size was the OHIP-14 for a pre-cision of 1.5 points of difference.”

As indicated by the referee, we have added the results of the normality tests. Although the Student's test is a robust test, and the sample size allows us to use the central limit theorem, In our case, this was not necessary since all the variables studied were normal for both the Shapiro-Wilk test (samples smaller than 50 individuals) and the Kolmogorov-Smirnov test (samples larger than 50 individuals).

“All outcome variables were adjusted for normality according to the Kolmogorov-Smirnov test as well as for the Shapiro-Wilk test.”

6.- We find the referrer's suggestion very interesting, and it is our intention to continue the study with respect to outcome variables and their possible applicability to patient satisfaction.

In this study, however, all clinical measures were surrogate variables to establish stage (I-IV) and grade (A, B or C) of periodontitis, which were considered endpoints for comparisons with respect to OHIP.

The biggest problem we encounter at present is that due to the variability of the examiners and the dispersion of stages and grades, the sample size is very large. We have therefore considered only the OHIP-14 results.

We have added the following paragraph explaining this situation:

“once patients have undergone non-surgical periodontal treatment, they suffer a significant dispersion. Some patients improve notably and only have to undergo maintenance (the minority). Other patients need to repeat the non-surgical treatment due to persistent inflammation or pockets larger than 4 mm with bleeding (the majority).

Finally, even if some of them are scheduled for a second non-surgical session, a variable percentage will end up needing surgical treatment, which implies longer re-evaluation times and a greater dispersion depending on the surgical technique used.”

7.- Thanks for the warning, the tables were out of order. We have reconfigured tables 1, 2 and 3 to make them clearer to read.

Table 1. Mean and standard deviation with respect to the 7 dimensions of OHIP-14sp pre- (OHIP-14sp1) and posttreatment (OHIP-14sp2) and the total score.

Variables

Mean (SD)

Mean (SD)

Matched Differences

OHIP-14sp1

OHIP-14sp2

Mean

SE

95% CI

p

Functional limitation

2.13 (1.76)

1.55 (1.71)

0.57

0.12

0.32

0.83

**

Physical pain

3.61 (1.89)

3.09 (1.70)

0.52

0.17

0.18

0.87

**

Psychological discomfort

2.73 (2.51)

2.25 (2.49)

0.47

0.16

0.14

0.80

**

Physical disability

0.84 (1.58)

0.60 (1.31)

0.23

0.11

0.01

0.46

*

Psychological disability

1.43 (1.82)

1.09 (1.65)

0.33

0.13

0.06

0.61

*

Social disability

0.64 (1.20)

0.45 (0.99)

0.18

0.08

0.02

0.34

*

Handicap

0.44 (1.11)

0.34 (1.02)

0.10

0.06

-0.03

0.23

0.13

Total OHIP-14sp

11.70 (8.38)

9.38 (8.16)

2.32

0.39

1.54

3.10

**

SD: standard deviation; SE: standard error; CI: confidence interval; *p < 0.05; **p < 0.01

Table 2. ANOVA of the significant dependent variables (post-treatment: 2) with respect to the grade of periodontal disease (A, C).

Dependent Variables

Grade

Grade

DM

SE

95% CI

p

Physical pain 2

A

B

-.943

.067

-1.95

-0.07

0.067

C

-1.401

.034

-2.69

-0.11

0.034*

DM: Difference of mean; SE: Standard error; CI: Confidence interval; *p < 0.05

Table 3: ANOVA of the significant dependent variables (pretreatment:1 and posttreatment: 2) with respect to the stage of periodontal disease (I, II, III, IV).

Dependent Variables

Stage

Stage

DM

SE

95% CI

p

Psychological discomfort1

III

IV

-1.55

0.71

-2.97

-0.14

*

Physical disability1

IV

I

1.27

0.62

0.03 

2,53

*

II

1.45

0.53

0.38 

2.53

**

III

0.94

0.44

0.06 

1.83

*

Psychological disability1

II

III

-1.39

0.54

-2.49

-0.30

*

Social disability1

IV

I

1.11

0.47

0.16 

2.06

*

II

1.13

0.40

0.32 

1.95

**

III

0.72

0.33

0.05 

1.39

*

Handicap1

IV

I

0.94

0.44

0.06 

1.82

*

II

0.87

0.37

0.12 

1.63

*

Psychological discomfort2

IV

I

2.00

1.00

0.01 

3.99

*

III

1.55

0.70

0.15 

2.97

*

Handicap2

IV

II

0.72

0.35

0.02  

1.42

*

Total OHIP-14sp1

II

IV

-6.18

2.89

-11.95

-0.42

*

DM: Difference of mean; SE: Standard error; CI: Confidence interval; *p < 0.05; **p < 0.01

8.- We agree with the referre and have simplified the discussion and eliminated pleonasms.

9.- We find the suggestion interesting. We have included possible lines of future research:

“Possible future lines of research could be oriented towards the conformation of a map of PROMs in the non-surgical treatment of Southern European countries. Another interesting line of research is the 6-month or 12-month follow-up on the degree of improvement observed by OHIP-14 with respect to the baseline score.

Likewise, it would be of great interest to continue the study with respect to the outcome variables (BL, PPD, BOP) and their possible relationship with respect to the degree of patient satisfaction.”

We would like to thank the referee for all the time and effort spent in supervising our work. We hope that this new corrected version will be considered appropriate for publication.

Sincerely yours.

Round 2

Reviewer 1 Report

The Reviewer comment was:

The greatest change in probing depth reduction and gain in clinical attachment occurs within 1–3 months post-scaling and root planing, although healing and maturation of the periodontium may occur over the following 9–12 months. (Cobb CM. Clinical significance of non-surgical periodontal therapy: an evidence-based perspective of scaling and root planing. Journal of clinical periodontology. 2002 May;29:22-32.)

The authors response:

“Although in advanced stages of periodontitis, periods of 6 to 8 weeks are recommended for re-evaluation, in our study, we tried to maintain a re-evaluation protocol of only 1 month for different reasons:

First, because of the length of the academic year, since treatments and reassessment are performed during the academic year, it is not possible to perform follow-ups longer than 1 month after the last non-surgical treatment. This is because students spend the first term acquiring basic skills.

NEW COMMENTS:

This is not a justification for maintaining a research bias, since these points or weaknesses must be assessed in the viability of the project when starting it. This is undoubtedly an important limitation and the justification of the "academic year" is not valid for strong research. The interpretation of the results and the conclusion are weakened by this fact.

If this point cannot be resolved and you want to keep this work as a first phase of a more in-depth study, it can be proposed as a pilot study or preliminary results.

Author Response

Reviewer 1

We accept the referee's suggestion. We have rethought the study as a pilot. This term has been added to the title.

The current title is: Periodontal Disease and Nonsurgical Periodontal Therapy on the OHRQoL of the Patient: A Pilot Study of Case Series”

We have also included the following sentence in the limitations of the study "For all these reasons, this study should be considered as a pilot study with preliminary results".

Thank you for your feedback

Reviewer 3 Report

The authors have addressed the major points raised. 

Author Response

Reviewer 3

Thank you for your feedback
